# Poly-(lactic-*co*-glycolic) Acid Nanoparticles Entrapping Pterostilbene for Targeting *Aspergillus* Section *Nigri*

**DOI:** 10.3390/molecules27175424

**Published:** 2022-08-25

**Authors:** Anastasia Orekhova, Cleofe Palocci, Laura Chronopoulou, Giulia De Angelis, Camilla Badiali, Valerio Petruccelli, Simone D’Angeli, Gabriella Pasqua, Giovanna Simonetti

**Affiliations:** 1Department of Public Health and Infectious Diseases, Sapienza University of Rome, P.le A. Moro 5, 00185 Rome, Italy; 2Department of Chemistry, Sapienza University of Rome, P.le A. Moro 5, 00185 Rome, Italy; 3Research Center for Applied Sciences to the Safeguard of Environment and Cultural Heritage (CIABC), P.le A. Moro 5, 00185 Rome, Italy; 4Department of Environmental Biology, Sapienza University of Rome, P.le A. Moro 5, 00185 Rome, Italy

**Keywords:** *Aspergillus* section *Nigri*, pterostilbene, PLGA-NPs, *Galleria mellonella*, biofilm

## Abstract

Poly-(lactic-*co*-glycolic) acid (PLGA) is a biodegradable, biosafe, and biocompatible copolymer. The *Aspergillus* section *Nigri* causes otomycosis localized in the external auditory canal. In this research, *Aspergillus brasiliensis*, a species belonging to the *Nigri* section, was tested. Coumarin 6 and pterostilbene loaded in poly-(lactic-*co*-glycolic) acid nanoparticles (PLGA-coumarin6-NPs and PLGA-PTB-NPs) were tested for fungal cell uptake and antifungal ability against *A. brasiliensis* biofilm, respectively. Moreover, the activity of PLGA-PTB-NPs in inhibiting the *A. brasiliensis* infection was tested using *Galleria mellonella* larvae. The results showed a fluorescence signal, after 50 nm PLGA-coumarin6-NPs treatment, inside *A. brasiliensis* hyphae and along the entire thickness of the biofilm matrix, which was indicative of an efficient NP uptake. Regarding antifungal activity, a reduction in *A. brasiliensis* biofilm formation and mature biofilm with PLGA-PTB-NPs has been demonstrated. Moreover, in vivo experiments showed a significant reduction in mortality of infected larvae after injection of PLGA-PTB-NPs compared to free PTB at the same concentration. In conclusion, the PLGA-NPs system can increase the bioavailability of PTB in *Aspergillus* section *Nigri* biofilm by overcoming the biofilm matrix barrier and delivering PTB to fungal cells.

## 1. Introduction

It is well known that despite the availability of several effective agents in the antifungal drug arena, their therapeutic outcome is less than optimal due to limitations related to drug physicochemical properties and toxicity. For instance, poor aqueous solubility and toxicity limit the formulation options and efficacy of several antifungal drugs [1]. On this basis, researchers have started exploring new opportunities for antifungal treatment, including novel antifungals and alternative approaches to treating fungal affections; for example, the use of nano vectors [2]. Moreover, recently, the development of nanotechnology and its applications in medical and health sciences has increased dramatically, allowing access to different kinds of nanoparticles (NPs), with well-defined active moieties to target human cells. A broad spectrum of drugs, such as small hydrophobic and hydrophilic drugs, as well as biological molecules, can be delivered in a controlled manner with NPs. NPs, ranging from 1 to 100 nm, can be easily employed as antifungal drug delivery vehicles. NPs have the possibility to enable closer contact with fungal cell membranes, thus facilitating their cellular uptake and controlled release of the drug within the cell environment [3,4,5]. Poly-(lactic-*co*-glycolic) acid (PLGA) is one of the most widely used and promising biopolymers for the development of drug delivery systems [6]. It is biodegradable and its degradation products, lactic acid and glycolic acid, are metabolized in the body via the Krebs cycle [7]. Therefore, PLGA systemic toxicity is negligible, and its use has been approved by the FDA and other regulating agencies. PLGA-based NPs and microparticles are currently being studied for the development of new drug-delivery systems for various drugs (i.e., chemotherapies, antiseptics, antioxidants), and some of them have already been approved by the FDA or are in clinical phase trials [6,8]. We previously reported the activity of PLGA-NPs in entrapping pterostilbene on *Candida albicans* and *Botrytis cinerea* [9,10]. To date, there are no publications on the activity of PLGA-NPs against *Aspergillus* biofilm. *Aspergillus* is a broad fungal genus comprising more than 300 different species, distributed ubiquitously worldwide. *Aspergillus* is a genus of filamentous fungi found in many habitats such as soil, air, water, and decaying plant material, and it can develop under a wide range of environmental conditions [11]. Species belonging to *Aspergillus* section *Nigri* have been difficult to classify due to their phenotypic similarities [12]. *Aspergillus* section *Nigri* includes species causing pulmonary aspergillosis and otomycosis in humans, as well as localized and disseminated diseases in domestic and wild animals. Otomycosis, the fungal infection caused by *Aspergillus* section *Nigri,* is localized in the external auditory canal, and less commonly in the middle ear [13,14]. Currently, the treatment options for otomycosis are still limited. The disease is hard to eradicate, and recurrence rates as high as 15% can be seen [15].

*Aspergillus* section *Nigri* is rarely differentiated at the species level when originating from human specimens. Some members of *Aspergillus* section *Nigri* (‘black aspergilli’) are *Aspergillus niger*, *Aspergillus welwitschiae, A. tubingensis*, and *A. brasiliensis*. *Aspergillus welwitschiae* is often collected from the external ear canal, whereas *A. tubingensis* and *A. niger* are predominant in respiratory samples [16]. *A. brasiliensis* is a species closely related to *Aspergillus niger* [17]. Moreover, *A. brasiliensis* (DSM 1988) is a reference microorganism, used to study the fungicidal effect of disinfectants and antiseptics (UNE EN 2019) [18].

Otomycosis is a biofilm-related infection. Resistance to antifungal treatments is also mediated by the development of *Aspergillus* biofilm, which provides temporary antifungal drug resistance and protects the pathogen in a hostile environment. The biofilm matrix reduces the entry and the diffusion of antifungal agents. New strategies against biofilms are needed. Pterostilbene (PTB) is a 3,5-dimethylated derivative of resveratrol that originates from several natural plant sources and that has shown strong activity against some fungal pathogens [9,19]. Recent literature studies have reported that PTB inhibits *C. albicans* biofilm [9,20]. The effect of PTB on fungal cells was related to the downregulation of the Ras/cAMP pathway and the ergosterol biosynthesis, which both contribute to the antibiofilm effect of PTB [21]. In the present study, the use of PTB loaded into NPs was evaluated with the aim of significantly improving drug performance [2]. In particular, PLGA-NPs entrapping PTB were employed to target *A. brasiliensis* section *Nigri* and its biofilm.

## 2. Results

### 2.1. NPs Preparation and Characterization

PLGA-NPs with different payloads were characterized by DLS, ζ-potential, and SEM measurements. Hydrodynamic diameter, PDI, ζ-potential, and morphology were not significantly affected by the entrapment of 6-coumarin or PTB inside PLGA-NPs. Typical samples presented an average diameter of 50 nm, with a PDI below 0.2 (Figure 1A), and had a ζ-potential of ~−25 mV. SEM analysis revealed spherical morphology for all preparations (Figure 1B).

PLGA-PTB-NPs had drug content of 0.375 mg of PTB/mg of PLGA, corresponding to an encapsulation efficiency of 75%. As previously reported [10] by the authors, the amount of PTB released in buffer solutions at different pH was low due to the poor water solubility of the drug, reaching the final value of 30 μg/mL within 5 days in plateau conditions.

### 2.2. Microscopic Observations of NP Uptake in Aspergillus Conidia, Mycelium, and Biofilm

In order to verify PLGA-NP uptake into *A. brasiliensis* conidia, mycelium, and biofilm, PLGA-NPs were loaded with the highly fluorescent probe coumarin 6, allowing them to be visualized under fluorescence microscopy. Results obtained with microscopic observations showed that 50 nm PLGA-coumarin6-NPs have the ability to penetrate conidia depending on their morphology. The conidia do not allow the internalization of NPs (Figure 2A, indicated by the yellow arrow). Only when the envelope breaks are NPs able to interact with the conidia capsule (Figure 2A,B red arrow). In the germ tube and hyphae, the fluorescence images report that 50 nm NPs were clearly visible up to 1 μm below the fungal wall (Figure 2C). The uptake of PLGA-coumarin6-NPs on biofilm was evident 60 min after administration (Figure 3A,B), the image showed that the 50 nm PLGA-NPs diffuse through the polysaccharide-derived extracellular matrix. No autofluorescence was detected from the conidia, mycelium, and biofilm alone (data not shown).

### 2.3. Antifungal Activity of PLGA-PTB-NPs

In vitro antifungal activity of free PTB and PLGA-PTB-NPs against biofilm in formation and preformed biofilm at different stages of formation (24 h and 28 h) has been determined by evaluating the in vitro metabolic activity of fungal cells. The results showed that after 24 h of incubation with PLGA-PTB-NPs, the PTB loaded into PLGA-NPs showed a significantly better activity compared to free PTB in all the experiments at a concentration of 20 µg/mL (Figure 4). Empty NPs did not show antifungal activity.

### 2.4. Activity of PLGA-PTB-NPs on a Model of Aspergillosis in G. mellonella

As previously reported, *G. mellonella* is a suitable model for testing the efficacy of antifungal agents against aspergillosis [22]. *G. mellonella* larvae were infected with conidial suspensions of *A. brasiliensis*. Mortality curves were used to calculate the lethal dose (data not shown). Assessment of the efficacy of PTB or PLGA-PTB-NP treatment was based on mortality in the lethal model. A dose-dependent reduction in mortality was observed after antifungal treatment with PTB and PLGA-PTB-NPs. PLGA-PTB-NPs were more effective than free PTB. The activity of PLGA-PTB-NPs was maximal at the highest concentration (Figure 5).

### 2.5. Toxicity on Galleria mellonella Larvae Model

PLGA-PTB-NPs were tested in an in vivo model by selecting the larvae of *Galleria mellonella* as a fungal in vivo model. *G. mellonella* was infected with *A. brasiliensis* (DSM 1988) conidia and treated with PLGA-PTB-NPs, PTB, and PLGA-NPs. To calculate the lethal dose, mortality curves were previously determined (data not shown). After inoculation with 4 × 10^4^ to 5 × 10^4^ conidia/larvae of *A. brasiliensis*, death was reported daily for 5 days. In the groups of untouched larvae and larvae inoculated with PBS, mortality was equal to 0%. Conversely, in the group of larvae infected with *A. brasiliensis*, the mortality rate was 45% at day 5 post-infection. Subsequently, using the same inoculum concentration, the larvae were inoculated with PLGA-PTB-NPs and PTB in a concentration range from 2.85 mg/kg to 0.089 mg/kg and from 1.35 mg/kg to 0.042 mg/kg, respectively. These results indicated a clear relationship between the concentrations of PLGA-PTB-NPs or free PTB and the mortality rate (Figure 5). The LD_50_ detected dose was more than 1.35 mg/kg for PTB and 2.855 mg/kg for PLGA, alone and in combination (Table 1).

## 3. Discussion

*Aspergillus* section *Nigri*, which include 26 species of black *Aspergillus*, are ubiquitous in the environment and in the hospital indoor environment. Clinically important species of *Aspergillus* section *Nigri*, such as *A. niger* and *A. tubingensis*, are the second most frequent agents that cause invasive aspergillosis and are the cause of otomycosis [23]. Among *Aspergillus* section *Nigri*, *A. brasiliensis* (DSM1988) has a conidial size of 3.5–4.5 μm, and growth and sporulation at 37 °C. Moreover, the growth on Czapek yeast autolysate agar with 5% NaCl is indicative of a close relationship with *A. niger* [17]. *A. brasiliensis* is used as a reference fungal species for the evaluation of the efficacy of compounds with antiseptic activity [18]. In this study, the well-characterized strain, *Aspergillus brasiliensis* (DSM1988) (ATCC16404), was chosen for use.

*Aspergillus* spp. are recognized as the fungi that form biofilms [24]. The negative consequences of biofilms are widely reported [14]. *Aspergillus* diseases are often associated with biofilm formation that increases host inflammation, rapid disease progression, and mortality. Biofilm is a microbial adaptation to the environment, resulting in antimicrobial resistance. Some *Aspergillus* species are resistant to antifungal treatment. The extracellular matrix, a defining feature of biofilm, is a complex mixture of biomacromolecules which contributes to reduced antimicrobial susceptibility.

The goal of this study was to investigate the antifungal activity of pterostilbene entrapped in poly(lactic-*co*-glycolic) acid nanoparticles (PLGA-NPs) on *A. brasiliensis* conidia, hyphae, and biofilm. Most studies have reported the antibiofilm effect of pure natural compounds included in nanosystems against *Candida* spp. biofilms, but only a few papers have reported the activity against *Aspergillus* spp. To eradicate biofilms, it is essential that NPs penetrate the matrix because it offers protection to pathogens, reducing cells’ susceptibility to antimicrobials [25]. However, knowledge of NPs–biofilm interactions is still limited. As previously reported, PLGA-NPs can penetrate fungal conidia and hyphae of species belonging to the *Aspergillus* genus. Patel and colleagues tested PLGA-NPs, loaded with coumarin 6, at 203 nm and 1206 nm and demonstrated that the uptake in *A. flavus* spores and mycelium depends on NP size. The smaller NPs tested (203 nm) were internalized more efficiently after 1 h of incubation than the bigger ones [26]. Muse and colleagues studied the uptake of PLGA-NPs covalently tagged with tetramethyl rhodamine isothiocyanate (TRITC), a red fluorescent compound (PLGA-TRITC), and PLGA-TRITC loaded with coumarin (double tagged), showing a clear surface association between NPs and *A. flavus* cells, while smaller NPs (30–50 nm) were internalized, allowing us to observe a red fluorescence inside the cells [27]. In general, for all cell types and organisms, the results suggest that smaller NPs are internalized more effectively by cells [28]. Moreover, PLGA-based NPs have been approved for many biomedical applications, such as delivery devices, and are considered safe for in vivo testing and applications. PLGA is a well-known random copolymer with physical and mechanical properties that can be easily tuned by altering the lactide to glycolate ratio, and by using nanoprecipitation techniques, it is possible to produce biodegradable nanoparticles with a controlled size and narrow size distribution [29]. In this study, we have shown that 50 nm PLGA-coumarin6-NPs can penetrate conidia only at a later germination stage; when the envelope breaks, conidia lose the pili layer and NPs can interact with the conidia capsule. Fang et al. have described conidia structure in detail. The authors reported that the surface roughness of conidia was approximately 33 nm [30]. Moreover, 50 nm NPs have been detected within the hyphae of the newly germinated conidia, at a distance of 1 μm below the fungal wall. PLGA-coumarin6-NP uptake by fungal biofilm was largely demonstrated and 50 nm PLGA-NPs diffused through the polysaccharide-derived extracellular matrix without any difficulty and remained entrapped even after rinsing with water. Regarding the antifungal activity, the anti-*Candida* activity of NPs -PTB- PLGA against *Candida albicans* biofilm was previously reported [9]. The increase in the activity of pterostilbene during the biofilm formation and after 24 h growth confirms our results, which were obtained by means of microscopy images. Moreover, recent studies have demonstrated that fungal virulence factors have similar and overlapping roles in mammalian and *G. mellonella* hosts, implying that *G. mellonella* studies can be transferred to mammals. *G. mellonella* are beneficial organisms for elucidating virulence factors, fungal signal or regulatory pathways, and the examination of antifungal pharmaceuticals [22,31]. In this study, PLGA-PTB-NPs proved to be more effective than free PTB in reducing, in a dose-dependent manner, the mortality of *G. mellonella*. Furthermore, the concentrations used in the in vivo tests were not toxic, as demonstrated in the evaluation of toxicity on *G. mellonella* larvae (LD_50_ more than 1.35 mg/kg for PTB and 2.855 mg/kg for PLGA alone and in combination). In conclusion, the in vitro and in vivo results have demonstrated that the antifungal compounds can be addressed by nanoparticles to reduce the infections caused by *A. brasiliensis*, section *Nigri*, biofilm.

## 4. Materials and Methods

### 4.1. Materials

Pterostilbene (PTB) was purchased from Chemodex (St. Gallen, Switzerland). Poly(D, L)-lactic-*co*-glycolic acid (PLGA, lactide: glycolide 50:50, MW 50 kDa), Coumarin 6 (98%), and all other chemicals and reagent-grade solvents were purchased from Sigma-Aldrich (St. Louis, MO, USA) and used as received.

### 4.2. Preparation and Characterization of PLGA-PTB-NPs and PLGA-coumarin6-NPs

PLGA-NPs loaded with either PTB (PLGA-PTB-NPs) or the fluorescent probe coumarin 6 (PLGA-coumarin6-NPs) were prepared based on our previous works, using a microfluidic reactor with a flow-focusing configuration through a nanoprecipitation mechanism [9,10,32]. Dynamic light scattering (DLS) and ζ-potential measurements were carried out using a NanoZetasizer (Malvern Instruments, Malvern, UK) to measure the mean hydrodynamic diameter of PLGA-NPs and their polydispersity index (PDI). The experimental conditions used are the following: a helium–neon laser operating at 633 nm, a fixed scattering angle of 173°, and a constant temperature (25.0 ± 0.1 °C). The measured autocorrelation functions of the scattered light intensity were analyzed using the CONTIN algorithm in order to obtain the decay time distributions, used to determine the distributions of the diffusion coefficients of the particles (D), converted into the distributions of the apparent hydrodynamic radii, R_H_, using the Stokes–Einstein relationship: R_H_ = k_B_T/6πηD (k_B_T = thermal energy; η = solvent viscosity). The electrophoretic mobility μ of particles, measured by combined laser Doppler velocimetry (LDV) and phase analysis light scattering (PALS), was converted into their ζ-potential using the Smoluchowski relation ζ = μ η/ε (ε=solvent permittivity).

Particle morphology was investigated by means of scanning electron microscopy (SEM) in both the secondary and the backscattered electron modes with an electron acceleration voltage of 20 keV, using an LEO 1450VP SEM microscope (ZEISS, Oberkochen, Germany). The quantitative analysis of PTB entrapped within PLGA-NPs was carried out using spectroscopic measurements. NP aqueous suspensions were ultra-centrifuged at a low temperature (14,000 rpm, 15 min, 4 °C) to recover NPs. The supernatant was discarded, the pellet was dissolved in acetone and analyzed by measuring the UV PTB absorbance at 313 nm, and the results were compared with a calibration curve within the concentration range between 0.002 and 0.01 mg/mL (R^2^ = 0.9822).

The encapsulation efficiency (EE) and loading capacity (LC) were calculated using the following equations:EE % = (total drug added-free non entrapped drug)/(total drug added) × 100(1)
LC % = (Amount of entrapped drug)/(total nanoparticle weight) × 100(2)

For studying the in vitro PTB release from PLGA-PTB-NPs, a fixed number of NPs were suspended in an acetate buffer at pH = 4.0 or in a PBS solution at pH = 7.4 and then incubated at 25 °C under magnetic stirring. At selected time intervals, fixed amounts of supernatant were collected and their PTB content was determined using the spectrophotometric method described above.

### 4.3. Fungal Strain and Culture Condition

*A. brasiliensis* (DSM 1988), formerly *A. niger* (DSM 1988), species *Aspergillus* section *Nigri* from the German Collection of Microorganisms (DSMZ, Braunschweig, Germany) was used as a reference strain in this study. *A. brasiliensis* (DSM 1988), the quality control strain, is used in preservative testing of pharmaceuticals, and in sterility testing. The strain was grown for five days on potato dextrose agar (Sigma Aldrich, St. Louis, MI, USA). The conidia were collected with phosphate-buffered saline and the concentration was determined with a Thoma counting chamber. RPMI medium (RPMI 1640 with L-glutamine, without bicarbonate) buffered to pH 7.0 with 0.165 M MOPS was used for antifungal tests.

### 4.4. Fungal Uptake of PLGA-coumarin6-NPs

The suspension of PLGA-coumarin6-NPs was added to *A. brasiliensis* (DSM 1988). *A. brasiliensis* conidia (1 × 10^5^ conidia/mL) were inoculated into 24 g/L of potato dextrose broth (PDB) and after 12 h and 24 h PLGA-coumarin6-NPs, at a final concentration of 0.1 mg/mL, were added. Fungal suspensions were placed on a microscope slide after 10 and 60 min. Mycelium, germ tube, and conidia were observed using an ApoTome fluorescence microscope. For PLGA-coumarin6-NPs uptake in *Aspergillus* biofilm, 1 × 10^5^ conidia/mL were cultured on glass microscope slides placed into Petri dishes containing PDB and incubated for 48 h. PLGA-coumarin6-NPs at a final concentration of 0.1 mg/mL were added to the biofilm. The biofilm was rinsed with sterile water and observed under an ApoTome fluorescence microscope after 60 min. The control of untreated conidia, mycelium and biofilm was observed to reveal any auto-fluorescence.

### 4.5. Microscopic Analysis

Conidia, mycelium, and biofilm treated with PLGA-coumarin6-NPs were observed, and images were acquired using an Axio Imager M2 fluorescence microscope (Zeiss, Wetzlar, Germany) motorized on the 3 axes by using a FITC filter (λ excitation BP 455-495 nm; λ emission BP 05-555 nm). The thickness of the sample provided a Z-stack image scan performed with an Axiocam 512 (Zeiss) monochromatic camera and ApoTome 2 (Zeiss) as a fringe projection module to eliminate the out-of-focus signal. Zen 2.5 (Zeiss) image analysis software was used to obtain single-plane images as Z-stack maximum projection.

### 4.6. In Vitro Antifungal Activity of PLGA-PTB-NPs against Biofilm Formation and Preformed Biofilm

In vitro antifungal activity against in formation and preformed biofilm was carried out as previously described [33]. *A. brasiliensis* conidia were inoculated with PLGA-PTB-NPs or free PTB and incubated in RPMI for 24 h and 48 h. Biofilm grown in 96 flat-well plates was inoculated with PLGA-PTB-NPs or free PTB.

After incubation for 24 h, the cells were washed and the metabolic activity, with 2,3-bis-(2-methoxy-4-nitro-5-sulfophenyl)-2H-tetrazolium-5-carboxamide (XTT) reduction assay, was evaluated. XTT-menadione was added, and after incubation, the optical density at 450 nm was measured. Each experiment was performed at least three times, in triplicate, on separate dates [33].

### 4.7. Toxicity of PLGA-PTB-NPs, Free PTB, and PLGA-NPs on Galleria mellonella Larvae Model

In vivo toxicity studies using the *G. mellonella* larvae were carried out as previously reported [34].

Larvae, of the sixth developmental stage of *G. mellonella* (Lepidoptera: Pyralidae, the Greater Wax Moth), (obtained from Blu Fish Rome) were stored in wood shavings in the dark at 18 °C before use. Larvae with color alterations (i.e., dark spots or apparent melanization) were excluded and those weighing 0.3–0.4 g were selected for experimental use. Larvae were injected with different PTB concentrations, free or entrapped within PLGA-NPs, or with empty PLGA-NPs. Controls included larvae injected with or without a sterile physiological saline solution. Survival was monitored over 120 h. Larvae death was monitored by visual inspection of the color (brown–dark brown) and lack of movement after touching them with forceps. Each experiment was performed in triplicate.

### 4.8. In Vivo Activity of PLGA-PTB-NPs

*G. mellonella* larvae were infected with *A. brasiliensis* (4 × 10^4^ to 5 × 10^4^ conidia/larvae), with or without different concentrations of PLGA-PTB-NPs or PTB. Survival was monitored over 120 h. Larvae death was monitored by visual inspection of the color (brown–dark brown) and lack of movement after touching them with forceps. Each experiment was performed in triplicate.

### 4.9. Statistical Analysis

The data were expressed as mean ± s.e.m. *p* < 0.05 was considered statistically significant. Statistical criteria, *p*, and other parameters are shown for each experiment. *G. mellonella* survival was displayed via Kaplan–Meier curves. The statistical data analysis was performed using the GraphPad Prism 8 software (GraphPad Software Inc., La Jolla, CA, USA).

## Figures and Tables

**Figure 1 molecules-27-05424-f001:**
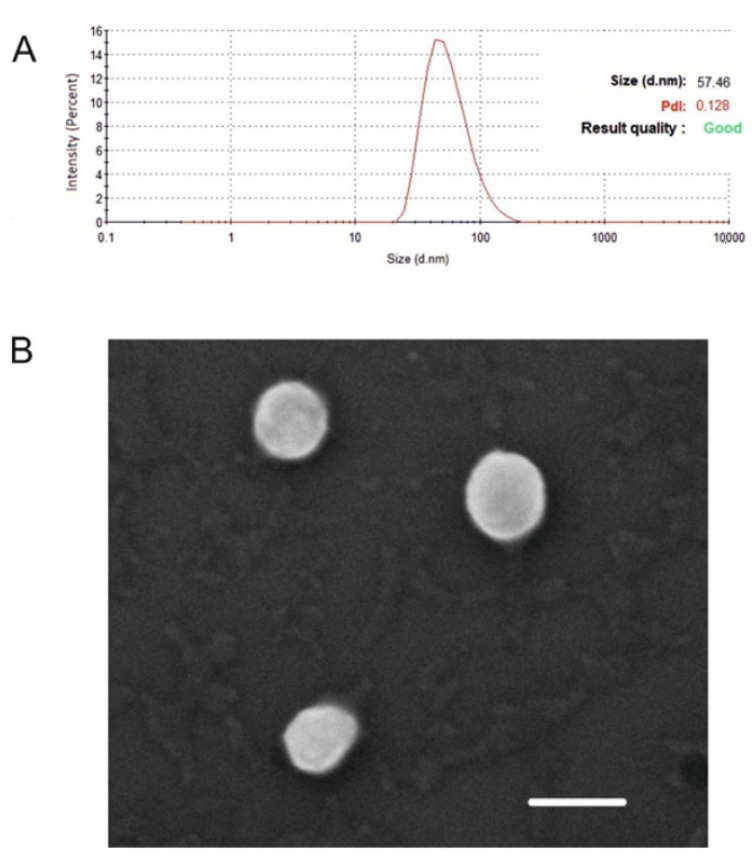
Size distribution by intensity and PDI of PLGA-PTB-NPs, as measured by DLS (**A**). SEM micrographs of PLGA-PTB-NPs (Scale bar: 100 nm) (**B**).

**Figure 2 molecules-27-05424-f002:**
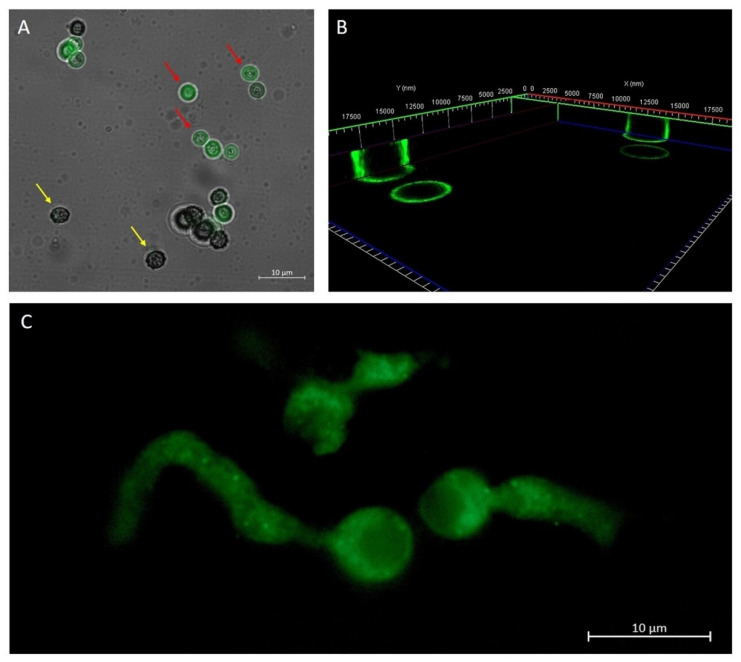
Overlap of the bright field image and the fluorescence image, which shows *A. brasiliensis* conidia treated for 10 min with 50 nm PLGA-coumarin6-NPs. In the first stage of conidia development (yellow arrow) the protective envelope did not allow interaction with NPs. In a later stage of conidia development (red arrow), when the envelope broke, fluorescence along the conidia capsule was observed (**A**). A 3D reconstruction of *A. brasiliensis* conidia treated with NPs for 10 min. The fluorescence signal was detected along the wall of the conidia (**B**). Fluorescence image of the hyphae of the newly germinated *A. brasiliensis* conidium treated with 50 nm PLGA-coumarin6 -NPs. The fluorescence signal inside *A. brasiliensis* hyphae is visible after 1 h of NPs administration (**C**).

**Figure 3 molecules-27-05424-f003:**
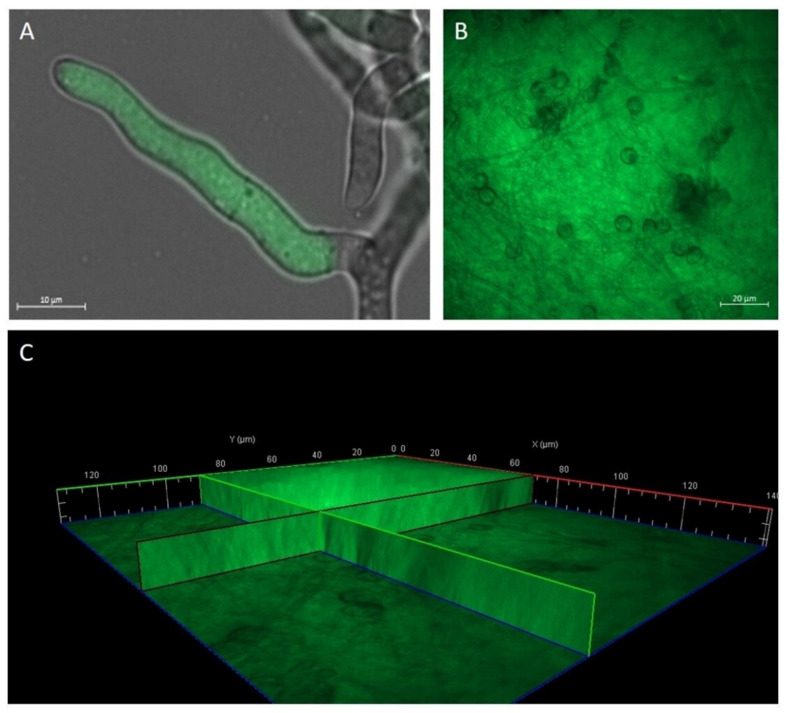
Observation of *A. brasiliensis* mycelium and biofilm treated with 50 nm PLGA-coumarin6-NPs. Overlap of the bright field image and the fluorescence image, which shows the localization of PLGA-coumarin6-NPs inside the fungal hypha (**A**). Presence of NPs within the biofilm (**B**). A 3D reconstruction of biofilm treated with NPs for 60 min. The fluorescence signal was detected along the entire thickness of the biofilm matrix (**C**).

**Figure 4 molecules-27-05424-f004:**
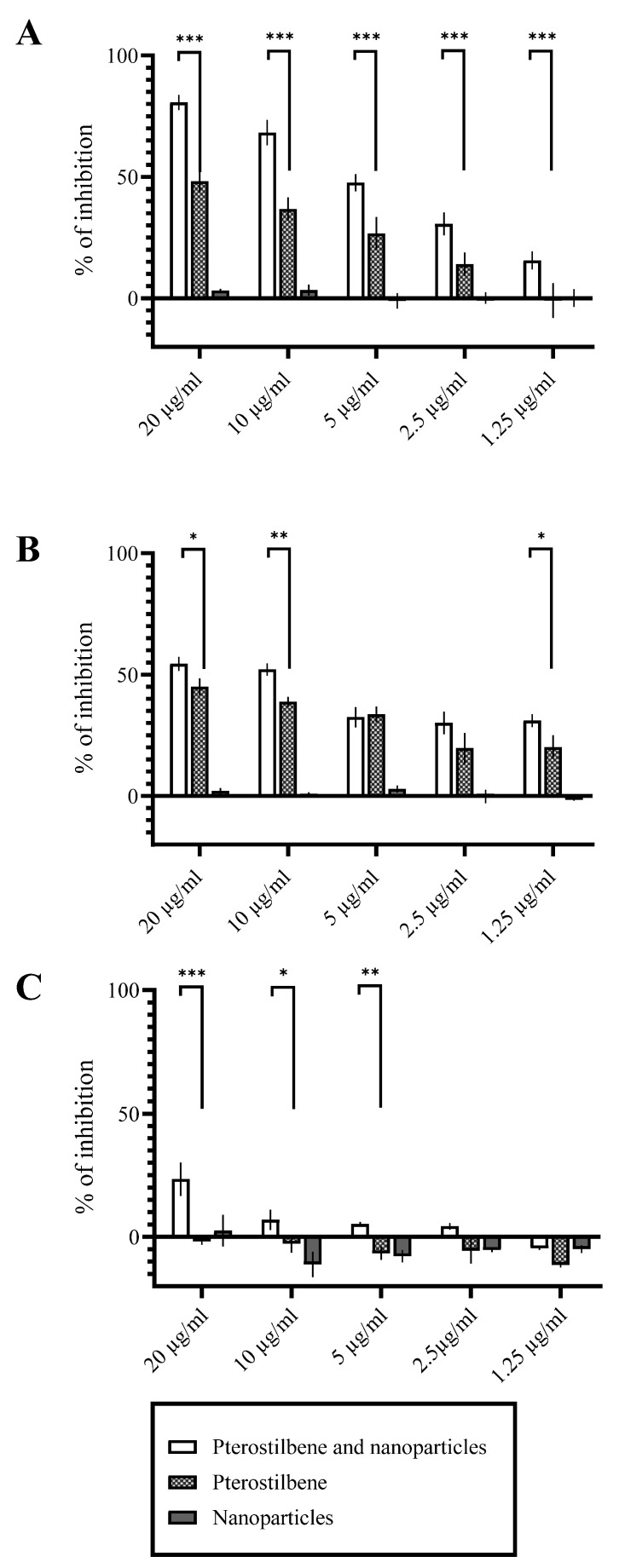
Activity of free PTB and PLGA-PTB-NPs against *A. brasiliensis* (DSM 1988). Activity of free PTB and PLGA-PTB-NPs against *A. brasiliensis* biofilm in formation, after 24 h of incubation (**A**). Activity of free PTB and PLGA-PTB-NPs against *A. brasiliensis* 24 h biofilm, after 24 h of incubation (**B**). Activity of free PTB and PLGA-PTB-NPs against 48 h biofilm, after 24 h of incubation (**C**). * *p* < 0.05 compared to the control; ** *p* < 0.01 compared to the control *** *p* < 0.001 compared to the control.

**Figure 5 molecules-27-05424-f005:**
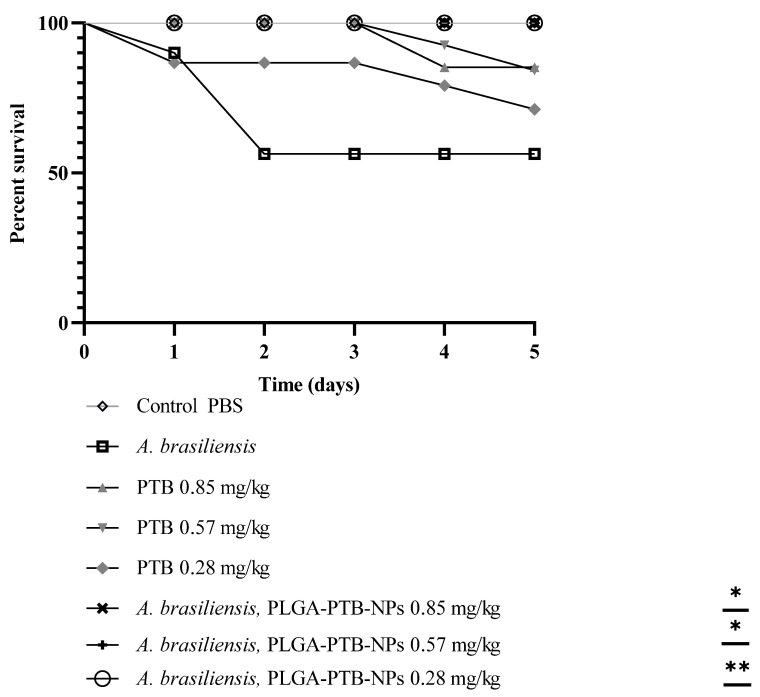
PLGA-PTB-NPs reduces *A. brasiliensis* virulence in *G. mellonella* model. Survival curves of *G. mellonella* larvae (*n* = 10/strain) infected via injection with 2 × 10^4^ conidia from *A. brasiliensis* with free PTB. Larvae were monitored for 5 days post-infection. Statistical significance relative to control was judged by the Kaplan–Meier followed by Mantel–Cox log-rank tests. At least three independent biological replicates were carried out for each experiment. * *p* < 0.05 compared to the free PTB and PLGA-PTB-NPs; ** *p* < 0.01 compared to the free PTB and PLGA-PTB-NPs.

**Table 1 molecules-27-05424-t001:** Survival of *G. mellonella* larvae following administration of PLGA-NPs, PTB, and PLGA-PTB-NPs by intra-hemocoel injection. Every experiment was conducted with 10 larvae for each group in triplicate. All values are the mean of three independent experiments.

Chemical	LD_50_ (mg/kg)	Solvent
PLGA-NPs	>2.85	H_2_O
PTB PLGA-PTB-NPs	>1.35 >2.85–1.35	100 H_2_O:1 DMSO 100 H_2_O:1 DMSO

## Data Availability

The data presented in this study are available in this article.

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
