# Peer review of "Poly-(lactic-co-glycolic) Acid Nanoparticles Entrapping Pterostilbene for Targeting Aspergillus Section Nigri"

_molecules, 2022, doi:10.3390/molecules27175424_

Round 1
Reviewer 1 Report
The paper is suitable for publication with minor corrections. The paper is well structured and written.
Minor revision
Since Aspergillus section Nigri contains 26 species, it is better to specify both in the Abstract and in the Introduction and Discussion sections that Aspergillus brasiliensis, section Nigri, was used and tested. In the introduction, the authors could specify the characteristics of this Aspergillus section and cite some of the main 26 species and why they chose to test A.brasiliensis and add the peculiarity of this Aspergillus if it has some. Furthermore, cumarin characteristics should be specified and why the authors chose to use this vegetal extract.
Add to the tables the species tested (A.brasiliensis), as the term Aspergillus is too generic.
Please explain how the conidia in M & M were obtained and how cumarin extract was prepared and conjugated with NP.
In the results section, cumarin alone data is missing.
(please add a new title in M&M about cumarin)
Author Response
Q1: Since Aspergillus section Nigri contains 26 species, it is better to specify both in the Abstract and in the Introduction and Discussion sections that Aspergillus brasiliensis, section Nigri, was used and tested. In the introduction, the authors could specify the characteristics of this Aspergillus section and cite some of the main 26 species and why they chose to test A.brasiliensis and add the peculiarity of this Aspergillus if it has some.
A1: Following the reviewer’s suggestion, Aspergillus brasiliensis, section Nigri has been specified throughout the whole manuscript. Moreover, we have added in both the introduction and discussion sections relevant information on Aspergillus section Niger brasiliensis with related references.
Q2: Furthermore, cumarin characteristics should be specified and why the authors chose to use this vegetal extract.
A2: As well known, fluorescent probes like coumarin 6 are widely used for the real-time detection, tracking and quantification of biomolecules in living cells as well as fluorescent markers for the entrapment efficiency evaluation of fluorescent nanoparticles (Yuansheng Sun et al., Chapter nineteen - Monitoring Protein Interactions in Living Cells with Fluorescence Lifetime Imaging Microscopy, Methods in Enzymology, Academic Press, Volume 504, 2012, Pages 371-391). Coumarin 6 is a commercially available chemical compound with high purity and it has been largely used by the authors in the past in many studies with the aim to track polymeric nanoparticles within cells (De Angelis et al., Scientific Reports 2022, 12, 7989; Simonetti et al., Molecules 2019, 24, 2070; Cacciotti et al., Nanotechnology 2018, 29, 285101; Palocci et al., Plant Cell Reports 2017, 36, 1917). Moreover, its utility comes from its capacity to absorb light of a specific wavelength (~450 nm), and emit light of a longer wavelength through fluorescence (~490 nm). In this study coumarin 6 was loaded into PLGA nanoparticles and used as a model molecule to evaluate nanoparticle uptake into conidia, mycelium and biofilm of Aspergillus brasiliensis. To improve the understanding of the experiments a sentence clarifying the role of coumarin 6 was added to the paragraph “Microscopic observations of NPs uptake in Aspergillus conidia, mycelium, and biofilm”.
Q3: Add to the tables the species tested (A.brasiliensis), as the term Aspergillus is too generic.
A3: Following the reviewer’s suggestion, the term Aspergillus was replaced by Aspergillus brasiliensis in figure 5 and in the graphical abstract.
Q4: Please explain how the conidia in M & M were obtained and how cumarin extract was prepared and conjugated with NP.
A4: Following the reviewer’s suggestion, we have added a sentence in the M&M section explaining how conidia were obtained. As explained above and reported in the materials section, coumarin 6 was purchased from Sigma Aldrich and used as received. PLGA NPs loaded with coumarin 6 were prepared by using an innovative microfluidic reactor with a flow-focusing configuration previously described by the authors (Chronopoulou et al., J. Nanopart. Res. 2014, 16, 2703). NPs formation occurs through a nanoprecipitation mechanism in the mixing channel and NPs can be recovered at its end. PLGA (2 mg mL-1) and the selected payload (coumarin 6, 40 μg mL-1) were dissolved in Acetone. The aqueous flow rate was 2000 µL min-1 while the organic phase flow rate was 100 μL min-1.
Q5: In the results section, cumarin alone data is missing.
A5: We thank the reviewer for the suggestion. We have previously published data on coumarin 6 entrapment in PLGA NPs (De Angelis et al., Scientific Reports 2022, 12, 7989; Simonetti et al., Molecules 2019, 24, 2070; Cacciotti et al., Nanotechnology 2018, 29, 285101; Palocci et al., Plant Cell Reports 2017, 36, 1917), so in this work we did not report the detailed characterization of PLGA-coumarin6-NPs, that can be found in our previous papers.
Reviewer 2 Report
The paper by Simonetti and coworkers describes the “Poly-(lactic-co-glycolic) acid nanoparticles entrapping pterostilbene for targeting Aspergillus section Nigri”. The authors presented the details in the manuscript in a good format. Nevertheless, there are still some issues needed to be addressed before publication in this journal.
Comments:
1. Please supplement the evidence to prove the entrapment of 6-coumarin or PTB inside PLGA-NPs (examples: FTIR, XRD, UV, etc.).
3. Please add the specific experimental process for the calculation of the weight of the loaded drug.
4. Please add the cell viability and cytotoxicity experimental work of PLGA NPs with payloads, if the author provides basic biocompatibility studies of PLGA NPs it would be more impact on the manuscript.
5. Please check for grammar corrections.
Author Response
Q1: Please supplement the evidence to prove the entrapment of 6-coumarin or PTB inside PLGA-NPs (examples: FTIR, XRD, UV, etc.).
A1: Regarding coumarin 6 entrapment we didn’t quantify its entrapment efficiency, that however must be almost 100%, since the aqueous supernatant we obtain after centrifugation of the NPs suspension is colorless, while coumarin 6, insoluble in water, typically confers a bright yellow color, even if present in small quantities. The quantitative analysis of pterostilbene within PLGA NPs was carried out by spectroscopic measurements, as reported in detail in previous papers (De Angelis et al., Scientific Reports 2022, 12, 7989). In particular, NPs aqueous suspensions were ultra-centrifuged at low temperature (4 °C) to recover NPs. The supernatant was discarded and the pellet was dissolved in DMSO and analyzed by measuring the UV absorbance at 313 nm, comparing the results with a calibration curve. The method used for pterostilbene quantification was linear within the concentration range between 0.002 and 0.01 mg mL-1 with R2=0.9822.
Q2: Please add the specific experimental process for the calculation of the weight of the loaded drug.
A2: The euqations used to calculate the encapsulation efficiency and the loading capacity have been added in the methods section.
Q4: Please add the cell viability and cytotoxicity experimental work of PLGA NPs with payloads, if the author provides basic biocompatibility studies of PLGA NPs it would be more impact on the manuscript.
A4: : The biocompatibility has been investigated by toxicological assays and safety test in Galleria mellonella. PLGA loaded with pterostilbene did not provoke toxic effects on the G. mellonella model, indicating that PLGA-PTB-NPs are biocompatible and safe. This model has been previously reported by many authors (Barros et al., Natural latex serum: characterization and biocompatibility assessment using Galleria mellonella as an alternative in vivo model. Journal of Biomaterials Science, Polymer Edition 2022, 33, 705; Allegra, E., Titball, R. W., Carter, J., & Champion, O. L. (2018). Galleria mellonella larvae allow the discrimination of toxic and non-toxic chemicals. Chemosphere 2018, 198, 469). Obviously further studies will be needed in the future for the clinical application of these NP systems.
The following sentence and reference have been added to the Materials and Methods section: In vivo toxicity studies using the G. mellonella larvae, were carried out as previously reported (Pandolfi, F., D'Acierno, F., Bortolami, M., De Vita, D., Gallo, F., De Meo, A., ... & Scipione, L. (2019). Searching for new agents active against Candida albicans biofilm: A series of indole derivatives, design, synthesis and biological evaluation. European Journal of Medicinal Chemistry, 165, 93-106).
Q5: Please check for grammar corrections.
A5: We have carefully revised the manuscript, correcting typos and e
Round 2
Reviewer 2 Report
It can be accepted in its present form.